# Acceptability of breast milk donor banking: A qualitative study among health workers in Greater Accra Regional Hospital, Ghana

**Fred Kafui Tende, Adanna Uloaku Nwameme📷, Philip Teg-Nefaah Tabong📷***

Department of Social and Behavioural Sciences, School of Public Health, College of Health Sciences, University of Ghana, Legon, Ghana

* ptabong@ug.edu.gh

**Data Availability Statement:** All data and related metadata underlying the findings reported can be obtained from the Ghana Health Service Ethics Review Committee on ethics.research@ghsmail.org.

## Abstract

Despite the compelling evidence demonstrating the immediate and long-term health advantages of prompt breastfeeding from birth, not all newborns are breastfed exclusively for a variety of reasons. As a result, the World Health Organization has made new recommendations for the adoption of breastmilk donor banks to make sure that children receive breastmilk when mothers are unable to produce it. In order to better understand how health professionals at the Greater Accra Regional Hospital, who would be leading the charge in putting this policy into practice, perceive and accept the practice of storing breastmilk, this study was conducted. At the Greater Accra Regional Hospital, 18 healthcare professionals were chosen using maximum variation purposive sampling procedures. They were made up of eleven midwives, a medical officer, six nurses, and two nutritionists and all participants were interviewed face to face using a semi structured interview guide. Data was transcribed verbatim and was analyzed using thematic analysis. Participants in the study admitted that they would be open to using or contributing to a bank of breastmilk. Participants also said that if safety precautions are taken, they would urge their customers to give breastmilk and recommend breastmilk from a breastmilk bank to them when the situation calls for it. Health professionals recommended that education be provided prior to the installation of breast milk donor banking to lessen or eliminate any misconceptions people may have about it. The concept of breastmilk donor banking was fairly accepted among health workers. Misconceptions about the safety of breast milk was the main concern. The results emphasize the necessity of greater stakeholder engagement and education prior to the implementation of this policy in order to boost acceptance and uptake.

## Introduction

Breast milk is the safest and most protective food for infants, and it's also the ideal choice for feeding premature and ill babies. For the first six months of life, breastmilk meets all nutritional needs [1]. Also, it shields the youngster from malnutrition and promotes healthy growth. Non-nutrient components of breast milk include immunoglobulins and lactoferrin,

 

**Funding:** The authors received no specific funding for this work.

**Competing interests:** The authors have declared that no competing interests exist.

which can help in intestinal maturation and adaptation as well as protection from viral and inflammatory illnesses [2]. The World Health Organization (WHO) advises only six months of exclusive breastfeeding. Up until the age of two, supplemental feeding is advised, and after that, it can continue for as long as the mother and child see fit [3]. Exclusive breastfeeding (EBF) is recommended to reduce infant mortality and improve health and cognitive development by preventing infection [4].

Only 42% of the 135 million babies born each year throughout the world receive only breast milk for the first hour, 38% of moms breastfeed exclusively for the first six months, and 58% of mothers exclusively breastfeed their kids until they are two years old [1]. Despite the fact that 98% of Ghanaian women claim to have breastfed their children at some time, infant feeding methods like exclusive breastfeeding and sufficient supplemental feeding are not frequently used [5]. USAID is working in Ghana to promote newborn and young child eating, including nursing practices, as one of many initiatives to lower the high rates of children who are too small for their age or stunted in the nation [6]. Historically and throughout cultures, informal breastfeeding sharing has been reported across cultures [2]. Peer-to-peer milk sharing is frequently done by mothers within their social circle [7]. Peer-to-peer milk sharing has the intention of providing human milk to infants whose parents are unable to provide any or enough milk, but this unregulated practice carries some risk because the milks are not subjected to any serological testing or screening, which means that diseases like Hepatitis B, syphilis, HIV, and other diseases may not be detected [8]. Human milk is the best food for all newborn humans. Nonetheless, there are several situations where the mother is unable or unwilling to breastfeed. A replacement is necessary in these circumstances [9].

The WHO recommends replacement feeding when breastfeeding, especially exclusive breastfeeding is not possible for medical reasons. Options for bridging the gap include donor breast milk from a human milk bank and formula feeding. Breast milk banks have been proposed as a means of preserving and donating breast milk to help feed neonates whose mothers are unable to breastfeed owing to several reasons [10]. At least 800,000 children worldwide receive donor human milk each year through breastmilk banking, according to a report by a virtual communication network of leaders in global milk banking from the year 2020 [11]. There are currently 248 Human Milk Banks spread across 26 European countries, according to the European Milk Bank Association (EMBA) [12]. However, the implementation of breastmilk banking has been slow in Africa [8].

According to estimates, only 42% of breastfeeding moms in Ghana continue to do so for the whole 20–23 months, whereas 43% of women exclusively breastfeed for 0–5 months [13]. Even though the Ghanaian government has put rules in place to encourage exclusive breastfeeding, there are still significant obstacles standing in the way of achieving the best possible outcomes for newborn health and wellbeing, necessitating the possibility of breast milk banking. In a previous study, 8% of mothers admitted to ever giving breastmilk to another mother without having the milk screened [14]. As a result, it's critical to investigate the WHO suggestion of donor banking for breastmilk, which offers the chance for the milk to be checked before use. Nonetheless, the support of important stakeholders like health workers is necessary for the implementation of this strategy. Hence, this study was conducted to explore breast milk donor bank acceptability among health workers in Greater Accra Region of Ghana.

## Methods and material

### Study area

The data for the study was collected among health workers at Greater Accra Regional Hospital. The hospital renders antenatal and delivery services to pregnant women and is a secondary

level government hospital supporting the leading referral centre in Korle-Bu Teaching Hospital, Accra [15], thereby positioning it favourably as the study setting study area. The hospital presently has a total staff strength of 654 comprising clinical and no-clinical staff [15].

## Ethical approval

The Ghana Health Service Ethics Review Committee (GHS-ERC 030/09/22) granted ethical approval for the study. Also, permission from the Greater Accra Regional Hospital's administration was sought in order to carry out the study. An informed consent document was signed by each study subject.

## Study design

The study relied on a narrative study design [16]. The research was designed to allow participants to express their understandings and perceptions of the meaning and experiences related to the breast milk bank because individuals and groups of people perceive a phenomenon differently [17]. The data for the study was collected among health workers at Greater Accra Regional Hospital. The hospital renders antenatal and delivery services to pregnant women and is a secondary level government hospital supporting the leading referral centre in Korle-Bu Teaching Hospital, Accra, thereby positioning it favourably as the study setting.

## Theoretical underpinnings of study

The acceptability framework created by Sekhon and colleagues [18] was used in the investigation. While planning, carrying out, and putting into practice wellness interventions, acceptance is a crucial factor to consider. Based on predicted or actual cognitive and emotional responses to the intervention, it is a complex term that reflects the degree to which those delivering or receiving healthcare interventions feel them to be acceptable [18]. The seven component constructs that make up the theoretical framework of acceptability (TFA) are affective attitude, burden, ethicality, intervention coherence, opportunity cost, perceived effectiveness, and self efficacy [18].

Affective attitude examines how a person feels about a particular intervention. The apparent effort needed to take part in an intervention is what is meant by the load. The degree to which the intervention fits well with the person's set of values is what constitutes ethicality. The degree to which participants comprehend the intervention and its mechanism of action is measured by its coherence. Opportunity costs examine whether gains, earnings, or values will be sacrificed while relying on an intervention directly or indirectly. The degree to which an intervention is considered as likely to fulfill its goals is covered by perceived effectiveness, and the participants' self-efficacy is examined by perceived effectiveness and self-efficacy, respectively [18]. These seven principles were combined in determining acceptability of breastmilk donor banking.

## Study population and sampling

Health professionals from Greater Accra Regional Hospital made up the study's target population. As stated earlier, the hospital has a total staff strength 654 comprising clinical and no-clinical staff [15]. In this investigation, maximum variation purposive sampling was used. To determine which hospital departments are involved in infant feeding, the researchers got in touch with the medical director of the facility. Based on this, the departments were approached to request participant nominations for projects that have a direct bearing on breastfeeding infants. A total of 18 health professionals, including 11 midwives, 4 nurses, 2 nutritionists, and

a medical doctor, participated in the study on the basis of data saturation. Before collecting data, consent forms were distributed to health personnel who wished to participate in the study.

### Inclusion and exclusion criteria

To qualify for inclusion in this study, the health worker should have been working in maternal and child health unit with at least one experience. Health workers who declined to give consent were excluded from the study.

### Data collection tool and procedure

An interview guide was used to conduct the in-depth interviews. Pre-testing of the interview guide was done among five health workers at Adabrak polyclinic in the Greater Accra region. The interview concentrated on the understanding and attitudes of health professionals on the appropriateness of breastmilk donor banking. One of the main issues was respondents' inclination or unwillingness to recommend families and potential families to give and/or use donated breastmilk, or to donate and/or use donated breastmilk to nourish their needed infants. Also, the data collector requested permission to record the interviews. Following each interview, participants' sociodemographic and professional information was collected on a separate form. All interviews were conducted face-to-face and in English by the lead author who has experience in qualitative data collection. As a method of participant validation, the interviewer outlined the main points after the interview [19]. An interview session takes 45 to 60 minutes to complete. All the interviews were conducted between 20th October 2022 and 2nd February, 2023.

### Data analysis

Using version 13 of QSR NVivo, data was examined. For this study, a thematic approach was used. The taped interviews were completely transcribed in order to guarantee credibility and objective results. The data was independently coded, and the transcript was cross-checked. Coding results were examined, and codes were examined for consistency, to assure intercoder concordance [20]. Themes and new issues were established when the codes were jointly reviewed. Themes and topics from the interview guide were initially used to analyze the data. An inductive technique was used to discover additional themes, sub-themes, and patterns in the data [21]. Finally, quotes from the study subjects were used to support the points made.

## Results

### Socio demographics of participants

The sociodemographic details of the participants are shown in Table 1. Of the eighteen (18) volunteers gathered for the study, eleven (11) were midwives. There were five general nurses, two nutritionists, and one medical doctor. Two (2) men and sixteen (16) women who responded to the survey were the respondents. The responders' average age was close to thirty (30) years. A postgraduate degree was the greatest level of education earned by the respondents, and a diploma was the lowest. Just two (2) of the respondents were Muslims but the majority, were Christians.

### Themes from data

A summary of the global, main and subthemes that emerged from the data is provided in Table 2.

**Table 1. Socio-demographic characteristics of study participants.**

| Characteristics | Number | Percentage (%) |
|---|---|---|
| *Gender Of Respondents* | | |
| Male | 16 | 88.89 |
| Female | 2 | 11.11 |
| *Profession Of Respondents* | | |
| Medical Officer | 1 | 5.56 |
| Midwife | 11 | 61.11 |
| General Nurse | 4 | 88.89 |
| Nutritionist | 2 | 11.11 |
| *Marital Status* | | |
| Married | 8 | 44.44 |
| Never Married | 10 | 55.56 |
| **Religion** | | |
| Christian | 16 | 88.89 |
| Islam | 2 | 11.11 |
| **Educational Attainment** | | |
| Diploma | 5 | 27.78 |
| Degree | 12 | 66.67 |
| Postgraduate | 1 | 5.56 |

## Knowledge on breastmilk donor banking

Breastmilk donor banking is described by healthcare professionals as the storing of expressed breastmilk for use in feeding other infants. According to a midwife's definition, it has also been used to feed infants whose moms are unable to breastfeed due to medical issues. While some healthcare professionals appeared to be unaware of the logic behind breastmilk donor banking, others were previously familiar with it. These health professionals said that their primary sources of information on breastmilk banks were the school, their peers, and the internet. As supported by these quotations, some definitions given by healthcare professionals regarding the banking of breastmilk were as follows:

> *"I learnt it's a bank just like the blood bank, where breastmilk is donated by various women, collected, screened (I mean it is passed through various investigations), packaged and stored under certain temperature and made readily available for mothers whom under certain conditions cannot breastfeed their babies."* (Midwife 1)

> *"It is the storing of donated breastmilk from mothers to feed children who have no opportunity to get breastmilk from their biological mothers."* (Midwife 2)

One midwife added that she learned about it while the COVID-19 outbreak was going on. The following quotation demonstrates this idea:

> *"To be honest I never knew of such a thing till the COVID-19 time when I learnt in the West, people were donating their breastmilk due to the shortage of formular feeds, that was when I knew you can even do such a thing."* (Midwife, participant 7)

In addition to some medical professionals declaring unequivocally that this is the first time they had heard of it, one midwife made a comparison to other methods of breastfeeding an infant, particularly when they are ill. The following passage supports my point:

**Table 2. Main themes and subthemes.**

| Global themes | Main themes | Sub themes |
|---|---|---|
| Knowledge about breast milk donor banking | Knowledge | • Knowledge on breastmilk donor banking |
| Perception of Health workers regarding breastfeeding and breast milk donor banking | Safety of breastmilk | • Fear of transmission of diseases such as HIV<br>• Fear of transmitting hereditary traits to infants |
| | Challenges and concern | • Low patronage<br>• Reluctance of mothers to donate<br>• Storage of breastmilk<br>• Risk of infections<br>• Funding<br>• Sustainability |
| | Benefits of breastmilk | • Breast milk bank is beneficial<br>• Reduction of infant mortality<br>• Protection against diseases |
| | Cultural, ethics and religious considerations | • Activities of some religious and traditional leaders<br>• Regulations of certain Christian churches<br>• The practice is ethical |
| | The use of infant formulas against breastmilk from a breastmilk bank (Pre NAN etc.) | • Cost<br>• Preparation for use<br>• Risks for infants<br>• Health benefits |
| Acceptability of breastmilk donor banking among health workers | Affective Attitude | • Excitement<br>• Happy |
| | Opportunity cost | • Loss of bond, attachment<br>• |
| | Intervention coherence | • Participants' understanding of its operation |
| | Perceived effectiveness | • Willingness to donate/ reluctance to donate breastmilk<br>• Willingness to use donated donor breastmilk for feeding their infant |
| | Things that need to be put in place before implementation | • Education<br>• Provision of qualified personnel<br>• Motivations<br>• Stakeholders involvement |

*"No, I don't know such a thing like that. What I know is like when your child is being admitted to the ICU (Intensive Care Unit), sometimes the mother can't go there and breastfeed so sometimes, they withdraw the milk into a container so that we feed the babies with those milk."* (Midwife, 5)

## Perception of health workers regarding breastmilk donor banking

**Benefits of breastmilk donor banking.** After the notion was conveyed to health personnel, they agreed that the introduction of breastmilk donor banking is a very good idea. Health professionals acknowledged that not every newborn has the opportunity to get breastmilk due to circumstances making it difficult for the mother to do so, and that the introduction of breastmilk banking will help solve this issue. The quotes that follow emphasize these viewpoints:

*"I think it's a good idea because as I said, sometimes when there is loss of life it becomes very difficult for anyone to just get up and breastfeed someone else's child but if something like this is at play, I feel it's going to be a very good idea and a very good source for the other babies who are lacking in that aspect to also benefit from the benefits of breastmilk."* (Midwife, 3)

*"It will benefit the baby and the mother at large. The baby will get all the nutrients required from breastmilk, it will increase the child's mental and general development and childhood disease and infections would be prevented."* (Midwife, 12)

There was widespread consensus that breast milk banks were beneficial since they will reduce premature mortality, according to the medical community. In light of the advantages, health practitioners generally had positive opinions about breastmilk banks. The quotes below serve to emphasize this:

*"I will say it will reduce infant mortality of premature babies whose mothers can't breastfeed or express their milk for use to feed the baby. It will also benefit orphans. It will benefit babies who wouldn't have mothers' milk to feed on they will also have the chance to be exclusively breastfed, mothers wouldn't have to worry about not been able to produce enough breastmilk for their kids, HIV infected mothers wouldn't have the problem to think about how they are going to breastfeed their infant."* (Nurse, 3)

*". . . It would reduce infant mortality because the introduction to breastmilk in an infant life will protect the child from minor illnesses and will boost the immune system of the child. Breastmilk is the best food for a newly born baby."* (Midwife, 14)

The practice of exclusive breastfeeding was a concern for health professionals, particularly those who were mothers, due to their line of work. They said that the necessary six months of exclusive breastfeeding could not be practiced during the three months of maternity leave and that the installation of a breastmilk bank would enable them to breastfeed their children entirely. The quotes that follow further explain this idea:

*". . .our maternity leave is not enough and sometimes expressing the milk too is time consuming for the mothers so if you are lazy or not dedicated, you cannot breastfeed so breastmilk bank will save us from all these. . ."* (Midwife, 1)

*"I did not do exclusive breastfeeding for the first two children, I did it for the last one. The first two because the annual leave is almost two months, and the maternity leave is three months, so I couldn't do it."* (Nurse, 2)

*"Sometimes the time isn't enough for us to breastfeed exclusively; even when it comes to expressing the breastmilk to feed your child, it also takes time so if you are not determined to breastfeed your child, then you might end up feeding your baby with the infant formulas which is also expensive and if not prepared well can harm your child."* (Nurse, 4)

**Culture, religion, and ethics of breastmilk donor banking.** Participants believed breastmilk donor banking was culturally appropriate. In addition, donor breastmilk banking was regarded as a moral practice. Health professionals believed that the practice was not novel and that it was appropriate for a community in Ghana. Health professionals noted that in the past, close relatives would wet nurse infants when the original mother was unable to do so. Health professionals emphasized that the introduction of breastmilk donor banking was an advanced form of wet nursing and that the practice had been discontinued owing to the risk of spreading diseases or infections. The quotes that follow demonstrate these ideas:

*"I would say it's not against our moral principle. It's a remedy to a problem so I don't see anything wrong with it."* (Nurse, 1)

"*It's already out there just that it's among families. My mother told me good stories about it; like you give birth, and you are no more, your mother or anyone that has given birth before can breast feed your child. You just need to stimulate the breast and the milk will come. This is just a means of formalizing it. If my sister gives birth and God forbids, she is no more, ah I will feed the child. People are doing it, I can assure you.*" (Nurse, 3)

"*In the olden days when someone delivers and dies or maybe is sick and there is a relative or a friend who has delivered, the person can breastfeed the child; but now fear of infections and all that, like STI. . ..*" (Nurse 2)

Christian and Muslim health professionals who were asked if their faith would prevent them from accepting breastmilk donor banks gave good responses. However, several medical professionals expressed worry that some Christian churches, whose names have been omitted, would not endorse it even though they are in favor. The following accounts lend credence to these opinions:

"*Religiously I think maybe the same group of people who have problem with the blood bank is the same people who will have issues with the breastmilk donor banking but personally my religion will not frown on it.*" (Nurse 2, Christian)

"*. . . my religion will tell me that anything that will benefit you it doesn't prevent it so religiously my religion wouldn't prevent me.*" (Nurse 3, Muslim)

"*I don't think I would have any problem with that personally and religiously as well because it's something that's going to benefit the child and not going to harm the child so religiously, I don't think there should be a problem with that.*" (Midwife 3, Muslim)

**Safety and risk of infections from contaminated breastmilk.** Despite the fact that the health professionals thought the intervention was effective, some people believed that given the important role breastmilk plays in developing the baby's immunization, it wasn't safe to use it to feed them. Infant feeding safety issues as well as the possibility that children could acquire genetic or inherited features from unknowing donors, including the danger of HIV infection, were highlighted. The quotes below demonstrate this position:

"*Genetically I don't know what is within that breastmilk production, you know, there might be cancer cells in that breastmilk that I am feeding because it's been produced by another mother and you know everything is made up of cells so it is cells that are generating all these things, so if it's generating something like that, I don't know whether there are traces of cancer cells in that person's family or something like a genetic disease or something. It's been screened o, but deep down it's not 100% safe.*" (Midwife 10)

"*It is a good intervention I will say but where I am reluctant or where I am holding back is, will it be safe? As in will it be hygienic and free from diseases like HIV? Are we having the resources to operate it in Ghana? It will be safe when it is screened for infections so that it doesn't cause harm to the infant.*" (Nurse 4)

Health professionals emphasized that there would be no cause for concern if only the donors and breastmilk were screened for illnesses to make it safe. The narratives that follow support this assertion:

*"So far as they will be screened for its safety, then it is good because there are people who are not healthy to do so but so far as it would be screened then it is a good initiative because breastmilk contains essential nutrients."* (Midwife, 12)

*"I was doubting because I don't think it would be safe for me to come and take breastmilk to feed my child in case of anything but if it undergoes processes and if all other things concerning its safety can be taken care of then there is nothing wrong about it."* (Midwife, 9)

A medical official who agreed that breastmilk should be checked for safety said that health professionals are also susceptible to making mistakes, such as wrongly declaring an HIV positive donor negative, which might have a significant detrimental impact on an unborn child. The following assertion demonstrates this:

**"***Fine it will go through some processes, mothers would be screened and all that but in the medical field too we tend to be negligent sometimes. You might mistake somebody's report for another's; let's say if a mother is retro positive and then the person goes to donate milk and we mix the report with another person's own thinking the mother is negative, what have you done to the baby? You have automatically infected the baby with HIV so when it comes to the medical side it is a no no. I also think it is not really necessary in this part of the world because we do well when it comes to breastfeeding.*"* (Medical Officer 1)

**Preference for donor breastmilk versus artificial formula.** Health professionals emphasized that in terms of health benefits, artificial formulas fall well short of breastmilk. Respondents claimed that poor preparation of formula had a detrimental impact on newborns. Health professionals believed that, despite the cost of saved breastmilk, it wouldn't be as expensive as infant formula. These respondents' opinions are illustrated by the quotes that follow:

*"Compared to formula feeds, breastmilk has a lot more nutrients than formula feeds. The feeds have some side effects of formula feedings including allergy and constipation. The cost of artificial formula is high. Even though donor milk will cost money, I don't think it will cost much more than artificial formulas."* (Midwife, 12)

*". . . breastmilk is highly nutritious than infant formulas; what role breastmilk plays cannot be compared to artificial formulas. It contains antibodies and other nutrients that help fight against diseases so health wise the breastmilk will be better in terms of this aspect and that's what we want to achieve."* (Midwife, 7)

*"With the artificial formulas, I don't think it is the best because it does not contain the necessary nutrients as compared to the breastmilk so with the artificial formulas, I myself I don't even like it when babies are giving artificial formulas."* (Midwife, 10)

There were two perspectives on how convenient donor breastmilk or baby formulae were. Regarding saving time, several of the responders thought donor breastmilk was more practical because it didn't require preparation time, unlike artificial formulas that needed boiling water and clean bowls and spoons to avoid contaminating the feed. Some experts took accessibility into account when deciding on the convenience. Formula feeds were thought to be more approachable in that regard. Breastmilk from a donor bank is more practical when decisions are made based on potential health advantages. The following narratives serve as examples of these viewpoints:

*"The breast milk or the breastmilk bank is convenient because you are not going to prepare it, you are just going to take and give; but with the artificial you are now going to prepare it. You need to look at the instructions, put water (boiled water hot cold, you know) and you either add more milk, less water or vice versa. But with the breast milk you just take it and you are giving it to the baby."* (Nurse 3)

"*With the convenience, the formula feeds will be convenient because it can be found in most shops to buy but with the donor breastmilk you will have to go to a breastmilk bank before you can get access to it.*" (Nurse 2)

*"There is no doubt that breastmilk is the best and it cannot be compared to formulas irrespective of its cost and convenience but as Ghanaians we will prefer something accessible and less expensive. So if the donor breastmilk is in existence and it's also going for a cost, mothers or families would prefer the formulas because for the formulas it's all over stores and you can get it anytime."* (Medical Officer 1)

**Challenges health workers anticipate with regards to breast milk donor banking.** The primary difficulty that health experts anticipated was the safety of donor breastmilk to feed newborns. They believed that people would continue to wonder if donated human milk would be safe to give to infants. Also, they expressed their uncertainty as to whether moms would feel the use of stored breastmilk to be suitable. The following statements provide examples of these viewpoints:

*"People will be concerned about the safety of the breastmilk; medically, had the breastmilk gone through screening? Is it safe? Would the baby get some form of, probably condition or diseases, should in case the donor have those conditions, like would there be some form of transfers that will affects the child's health?"* (Nutritionist, 1)

*"I think the concern people might raise more is you not knowing the person the breastmilk is from, people not being sure. You said you are screening, are you sure you screened it? And then the conditions, the hygienic condition; I think these are the things people would raise concerns about."* (Midwife, 2)

*"People will also be concerned about the safety of it whether it has really undergone some level of screening to make it safe. So, if they are 100% sure that the breastmilk does not contain any infections and that when given to my baby it wouldn't harm him or her."* (Midwife, 10)

Health professionals also expected that mothers could be reluctant to give breastmilk to a bank, which might lead to shortages and undermine the bank's success. As we don't have enough competent workers, some health professionals expressed concern that its safety would be at risk. The quotes that follow emphasize these ideas:

"*The operation of donor milk banking will depend on mothers donating breastmilk to the bank so if in future mothers feel reluctant to donate, the operation wouldn't achieve its purpose."* (Midwife 8)

*"I feel it would be hard to get donors because this is not the need now in our hospitals and maybe certain cultural beliefs will also hinder people from patronizing it. I also feel its safety would be compromised because we don't have the personnel, I mean the technical know-how to handle a breastmilk bank. Even when it's been operated, the bond between mother and child might be lost."* (Medical Officer, 1)

*"There can be shortage of breastmilk when it becomes rooted, and everybody is purchasing, and we are not getting people to replace the breast milk."* (Midwife,1)

Some medical professionals were also concerned about how breastmilk is stored and how long it may be kept in the breast bank before becoming bad. The following quotations support this claim:

**"**. . .*how long will be breastmilk be stored because even if we have donors and people don't come for the breastmilk the milk might go waste."* (Midwife 8)

*"How long can it be stored without the milk going bad? If mothers are given enough breastmilk to take home, can our normal refrigerator store it? If yes, at what temperature? Will mothers avail themselves to donate at a breastmilk bank?"* (Midwife, 6)

Health professionals also predicted that a difficulty to the complete implementation of breastmilk banking in Ghana would be funding and the sustainability of breast banks. Some examples support this viewpoint:

*"I think that we do not have the resources unless we are getting the support from maybe international bodies. Funding can be a problem and after getting it done sustainability can also be a problem."* (Midwife, 6)

*"Some of the challenges will be the source of funding. Looking at our current economy, I am wondering if this would be possible. Aside from the funding, people will still be worried about whether it is safe for feeding infants. Will mothers come out in their numbers to donate?"* (Midwife, 3)

To ensure uptake of the breastmilk donor banking, interviewees suggested the cost of donated milk should be made less costly than the artificial formulas. It was recommended that dedicated funding sources should be ascertained before the role out of such a policy. The following illustrate these points:

"Number one I will say is funding to carry out such an intervention. Funding can be an issue if they don't dedicate enough funding for it" Nurse (Nurse 3)

**Acceptance of breastmilk donor banking.** Health professionals who could explain the advantages to potential clients were generally in favor of the idea of breastmilk donor banking. Participants were excited about the idea of breastmilk banking and gushed about how it would help moms in their quest to feed their babies by saving them time and money. The sentiment expressed by medical professionals concerning the creation of breastmilk donor banks is exemplified by the quotes below:

*"I'm very excited about this. I feel is a good plan, it should come, and it should stay. I am positive about this- I hope it works out."* (Midwife, 3)

**"***I will be really excited to see it being implemented here in this hospital. As I said, it's going to help a lot because there are instances where people must go outside to go and feed their baby. So, with this it's going to help these mothers a lot; it will save money and time."* (Nurse, 3)

The study's findings also demonstrated that establishing a breast milk bank in Ghana would be successful in the long run. Health professionals predicted that it wouldn't be simple to maintain it at first, but that it would eventually succeed.

*"It would work as long as the organizations are willing to support, and mothers are willing to offer themselves as point of donors it will work but it will come with its own resistance. Even if it doesn't work at the first trial and those gaps that made it not to work are fixed then with time it would work."* (Nutritionist, 1)

*"It will work. Starting something is difficult but if people notice its benefit, they will accept it."* (Midwife, 7)

*". . . there are some people who will accept it and there are some who will not accept it. We must start and see the way forward because I believe that interventions are not meant for everyone because there are some people who will be in dire need of it and those people will accept and patronize it."* (Midwife, 12)

When the necessity arose, health professionals expressed their willingness to donate and/or use donated breastmilk to feed their infants. Health professionals were also willing to advise individuals in need to use donated breastmilk to feed their infants when necessary and to urge moms who have extra milk to give it to a breastmilk bank. The quotes below demonstrate these ideas:

*"If I'm in the place to donate fine and if there is someone who can donate too, I think I will advise because if it helps another child, why not?"* (Midwife, 9)

*"Yes, I will. Once it's going to help the baby and the mother, I will, and I will also advice someone to do it."* (Midwife, 4)

*"For me, at the moment I have finish giving birth so I don't think I will have any, but I'm willing to advice other women or colleagues to participate if they have enough to give."* (Midwife, 8)

While some healthcare professionals held the opinion that if they had an excess supply of breastmilk, they would donate it to a breast bank and advise women without medical conditions to do the same, others emphasized that they would only do so if the security of the breastmilk banking can be ensured.

*"If there are reports that the safety of it has been compromised, I wouldn't keep recommending it to people. If the safety of it is compromised, then there is no point of buying something that is not worth the quality."* (Nutritionist, 1)

*"If the donor is ill and doesn't have the strength to donate, I will not advise; aside that I don't think something will prevent me from recommending it."* (Nurse, 3)

**Pre-implementation preparations.** All of the healthcare professionals agreed that the key strategies for making this intervention successful were education and training. From the planning stage through the implementation stage, participants recommended continuous education. Health professionals believed that the only way to dispel myths and misconceptions about breast milk banks was via education. Before beginning this intervention, health professionals recognized the need to educate stakeholders, moms, couples, and the general public. Stakeholders should be involved at the planning stage, according to health professionals. The following quotations serve as examples of these viewpoints:

*". . .the public should be educated about it. It is very necessary because it's going to eliminate their negative perception about it especially about the safety of it."* (Midwife, 1)

*"There should be consistent education on it, through the media, opinion leaders, churches, the community should be involved, and this will make it easier for it to be implemented. The community should also be assured of the safety of the donor milk."* (Midwife, 5)

*"There should be qualified personnel that would be in charge of it, there should be intensive education to the public concerning its operation and the processes breastmilk would go through because it is something new."* (Midwife, 12)

Health professionals also suggested ways to reward moms who agree to give breastmilk to a bank. They believed that rewards in the form of incentives could inspire or uplift the spirits of moms who have extra breast milk and want to donate it to a breastmilk bank. They viewed this drive as a tool to persuade moms with extra milk to voluntarily offer it to a breastmilk bank as a donation. The quotes below demonstrate these ideas:

*If the nearest donor facility is at Korle Bu and I am here at Ridge I will not pick a car and go so the government, WHO, UNICEF and other bodies should heavily subsidize it and maybe compensate mothers who donate their milk."* (Nurse, 3)

*"I wouldn't donate if I won't be given any supplement to replace the energy I used to donate [laughs]. Yes, at least donors should be compensated. I think with this, mothers would feel encouraged to donate to a breast bank."* (Midwife, 1)

Finally, health professionals underlined the need to involve stakeholders in the development of a policy to direct and regulate the nation's breast milk bank before implementation, as seen in the following example:

*"There is a need for stakeholders' engagements before implementation. This would be an initial good step. In addition to that we need to develop a policy to guide the process. Otherwise, you may several people setting up breastmilk banks because fo monetary gains"* (Medical Officer 1)

## Discussion

This qualitative study looked at how well-liked breast milk donor banking is among medical professionals. The study makes it clear that some health professionals initially did not think this study was necessary because they knew little to nothing about breastmilk donor banks. Hence, there was very little initial acceptance among these health professionals; nevertheless, after being explained the basic idea behind breastmilk donor banking, the majority of the professionals displayed a favorable attitude. It sent a message that someone might dismiss an intervention if they have little or no awareness about it. Health professionals who were familiar with the practice of wet nursing also acquired a favorable perspective on the activities of breastmilk donor banking. This was due to the fact that these health professionals equated breastfeeding a newborn with the milk of a close friend or family with the practice of breastmilk donor banking. Health professionals who said they were aware of and knowledgeable with breastmilk donor banking were able to define it correctly. Colleagues and educational institutions were cited as the primary sources of information. The study found that understanding breast milk donor banking has a beneficial impact on people's acceptance of its practice. This outcome from the current study is supported by research done years earlier. A Brazilian study found that donating human milk for primary healthcare was strongly correlated with knowledge of milk expression [22]. According to a related study by Chagwena et al., having in-depth knowledge of breastmilk banks was associated with acceptance of donor human milk banking

[23]. The main obstacles preventing postpartum women from donating or accepting donor milk, according to Zhang et al., are a lack of knowledge regarding breastmilk donor banks and its safety [24]. This calls for sustained health education and community engagement before implementation as suggested by respondents in this study. Education on the use of breastmilk donor banking could be incorporated into the training curriculum of health workers and in-service training as they have a critical role to play in its implementation.

The study also revealed that health professionals believed the establishment of breastmilk banks in Ghanaian hospitals was an excellent concept and a great way for newborns who are lacking in that area to still benefit from breastmilk. According to the study, feeding needy children donor breastmilk will lower infant mortality and morbidity because the babies will have ready access to food, reducing the risk of infections. As has been reported in earlier study, child is more likely to survive when immediate exclusive breastfeeding was practiced for up to 6 months [25]. It was believed that giving a needy baby breastmilk from a bank would enhance the percentage of exclusive breastfeeding, hence preventing disease and mortality, because health practitioners emphasized how healthy breastmilk is. Similar reasons were adduced in a study among health workers in Zimbabwe [22]. These benefits could be highlighted in providing education on breastmilk donor banking.

According to the study, health professionals believe that obtaining breastmilk from a bank is preferable than using infant formula because the latter exposes babies to diseases when it is not produced hygienically. Yet, it was found that moms who couldn't afford to buy breastmilk from a breastmilk bank would choose to buy infant formula instead if the price of doing so rose. Cost of donor breastmilk has been reported as a barrier in Italy [26]. It is therefore imperative to find a sustainable way of reducing the cost. The possibility of adding it to the benefits of national health insurance could be explored.

The responders emphasized multiple times the importance of breastmilk for a baby's growth and development, and by implication, the importance of donated breastmilk. The impact of mothers' milk on preterm infants' neurodevelopment was previously demonstrated in a study conducted in the United States [27]. An associated British study found that donor breastmilk is better accepted and less likely to result in necrotizing enterocolitis in premature neonates [27]. Yet, different worries and opinions on the security of breastmilk from a bank were voiced. The main perceived barrier to the acceptance of breastmilk donor banking among health professionals was the fear of contagious illnesses. There were rumors that an HIV-positive mother could be mistaken for a donor who had tested negative, which could ultimately have an impact on safety of breastmilk. Health education to community on breastmilk banking should that all donated milk is screened before storage and use.

So, one of the most important tools in eradicating the worry that babies may catch diseases or infections was trust in the breastmilk banking system. In a research to investigate the acceptability of donated breast milk in a resource-constrained South African setting, lack of trust in HIV testing as well as healthcare providers and employees was raised, and this may alter attitudes toward breast milk donation [28]. In this study, misconceptions about the difficulty of feeding infants with donated breastmilk were also noted, including the transmission of infections and the transmission of inherited features or character. Thus, these medical professionals advocated the usage of infant formulae over the strategy of donor breastmilk banking. It was discovered that wet breastfeeding, which was practiced in the past, is no longer a popular practice due to concern over illnesses like HIV. Because of false beliefs regarding the safety of breastmilk, a comparable study carried out in Ethiopia by Gelano et al. revealed that the acceptance of breast milk donor banks and its use for feeding newborns was relatively low [2].

Results also showed that people's reluctance to give breastmilk or accept its use for newborn feeding instead of choosing formula was significantly influenced by religious groups' or

churches' activities. Some medical specialists believed that breastmilk donor banks would be frowned upon by various religious groups and churches. Hence, it was assumed that the behavior would appear improper to members of such churches. In a related study carried out in Ethiopia by Gelano et al., it was also discovered that mothers' reluctance to contribute to a breastmilk bank was partly motivated by their religious beliefs [2]. Notwithstanding the expected difficulties in implementing breastmilk banks in our hospitals, it was determined that the advantages of doing so would outweigh the difficulties. The study's findings also demonstrated that cultural contexts or conventions did not truly provide a barrier to using donated breastmilk to feed newborns. This resulted from the fact that wet nursing is currently practiced in most cultures or previously was. This calls for stakeholder involvement prior to the policy's implementation in Ghana.

Health professionals praised the value of depositing breastmilk donors and indicated their pleasure and approval with how breastmilk banks operate. The results demonstrate that health professionals were prepared to use donated breastmilk when necessary and were also prepared to suggest it to families in need due to the significance connected to it. For the most part, respondents said they would be comfortable delivering or recommending it. The desire to assist newborns in need served as the main driving force behind breastmilk donation. The acceptance of the intervention was also aided by health personnel' comprehension of the procedure and the issues the intervention typically resolves. It was clear that the safety of breastmilk donor banking and the understanding of health professionals were key factors in its acceptance. Breast milk donor banking was observed to be acceptable among medical professionals in a comparable study conducted in Zimbabwe [23]. It was discovered that some health professionals 31% said they would give their child donated breastmilk, while the bulk of professionals 56% agreed to advise their customers to donate breastmilk to a bank. Mothers expressed support and desire to give their breastmilk to breastmilk banks in a different study conducted in South Australia, providing the process is simple and quick [28]. Also, studies showed that moms of newborns who were premature or sickly would use a human milk bank if they were confident the milk was suitable and secure for their kids. To combat the false beliefs people, have about safety, education is a vital tool. This recommendation was in line with a number of studies that were done on people's desire to use breastmilk from a breastmilk bank and their willingness to contribute breastmilk to one. Public education on breastmilk banks is required, according to a study by Zhang et al. [24]. It concluded that information should be shared in the early stages of its establishment. Thus, efforts should be made to improve mothers' awareness of the benefits of breastmilk and nursing.

### Limitations of the study

Even though this study provides useful insights on breastmilk donor banking, it is important to note a few limitations. One weakness in qualitative studies is the inability to generalize the findings [17]. Nonetheless, we employed maximum variation sampling technique involving participants from different health professional groups to strengthen the findings of the study while increasing the credibility, dependability, and trustworthiness [29] of the evidence from the study. This study was also conducted in one Hospital, it would be important in future to expand such as study to cover different levels of health care delivery.

### Conclusions

Before the study, health professionals had little knowledge about the banking of breastmilk. There were misconceptions regarding its safety, with the biggest obstacle that can affect acceptability being the fear of contracting HIV. Despite this, medical professionals were upbeat and

optimistic about the importance of breastmilk banking and its effectiveness in reducing infant mortality and morbidity. When necessary, health professionals were willing to create awareness campaigns about breastmilk banking, give or use recipient families' donated breastmilk, and convince other families to do the same. The safety of breastmilk donor banking must be made known to women, parents, and the general public in order for it to be successfully implemented. Thus, breastmilk donor banking was acknowledged by healthcare professionals at Greater Accra Regional Hospital. In addition to the policy and practice recommendations, the study has provided pointers and themes that could be used to develop a data collection tool for a quantitative study in future.

## Acknowledgments

The authors wish to thank all the study participants who shared their views with the study. Team on the topic.

## Author Contributions

**Conceptualization:** Fred Kafui Tende, Adanna Uloaku Nwameme, Philip Teg-Nefaah Tabong.

**Data curation:** Fred Kafui Tende.

**Formal analysis:** Fred Kafui Tende.

**Investigation:** Fred Kafui Tende.

**Methodology:** Adanna Uloaku Nwameme, Philip Teg-Nefaah Tabong.

**Software:** Philip Teg-Nefaah Tabong.

**Supervision:** Adanna Uloaku Nwameme, Philip Teg-Nefaah Tabong.

**Validation:** Adanna Uloaku Nwameme.

**Writing – original draft:** Fred Kafui Tende.

**Writing – review & editing:** Adanna Uloaku Nwameme, Philip Teg-Nefaah Tabong.

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
