## [Decision Letter · Decision Letter 0]

13 Jun 2023

PGPH-D-23-00571

Acceptability of Breast Milk Donor Banking: A qualitative study among Health Workers in Greater Accra Regional Hospital, Ghana

Dear Authors,

Thank you for submitting your manuscript to PLOS Global Public Health. After careful consideration, we feel that it has merit but does not fully meet PLOS Global Public Health’s publication criteria as it currently stands. Therefore, we invite you to submit a revised version of the manuscript that addresses the points raised during the review process.

We look forward to receiving your revised manuscript.

Kind regards,

Shela Hirani, PhD, IBCLC, RN

Academic Editor

Journal Requirements:

Additional Editor Comments (if provided):

Reviewers' comments:

Reviewer's Responses to Questions

**Comments to the Author**

1. Does this manuscript meet PLOS Global Public Health’s publication criteria? Is the manuscript technically sound, and do the data support the conclusions? The manuscript must describe methodologically and ethically rigorous research with conclusions that are appropriately drawn based on the data presented.

Reviewer #1: Yes

Reviewer #2: Partly

2. Has the statistical analysis been performed appropriately and rigorously?

Reviewer #1: N/A

Reviewer #2: N/A

3. Have the authors made all data underlying the findings in their manuscript fully available (please refer to the Data Availability Statement at the start of the manuscript PDF file)?

Reviewer #1: No

Reviewer #2: No

4. Is the manuscript presented in an intelligible fashion and written in standard English?

Reviewer #1: Yes

Reviewer #2: Yes

5. Review Comments to the Author

Reviewer #1: Congratulations on your paper! You have done a great work! It is essential to understand health workers knowledge about breast milk donor bank process before implementing this kind of health facility.

Still, I have some comments to share:

Introduction: has relevant information and guides readers to comprehend your objetive.

Methods: concise, detailed, well structured. Recording interviews is a clever way to review informatioon and clarify any misunderstanding.

Results: Presenting your findings in sections makes it easier for readers to analyze them.

Data presented in Table 1 diverges from data in text.

Discussion: your results corroborates the findings presented in other articles.

You should consider making your data available through an easier access link, such as a support information link.

Reviewer #2: Abstract

Concise and clear. p.2

Introduction

Clarify the sentence ʿIf possible and safe, WHO advises........., may be hamperedʾ. line 81 and 82 . p. 4

Reference the texts on line 92-97. p. 4

Methods

Study area

Put the following information ʽThe data for the study was collected among health workers at Greater Accra Regional Hospital. The hospital renders antenatal and delivery services to pregnant women and is a secondary level government hospital supporting the leading referral centre in Korle-Bu Teaching Hospital, Accra, thereby positioning it favourably as the study settingʼ under study site and reference it. line 115-119. Also include information on the total number of health workers in the facility P. 5

Population

Correct the statement ʽAccra's Greater Accra Regional Hospitalʼ on line 114 P.6

Also include information on the total population of health workers in the facility

Include the eligibility criteria

Data collection tool and procedure

How did you ensured reliability and validity of the interview guide?

What language was used to conduct the interviews? Which type of interview was conducted? P. 6

Results

Sociodemographic characteristics

Clarify the statement ‘There were five nurses’, line 175 p. 8

Perception of health workers regarding breastmilk donor banking

Clarify the statement ʽBenefits of breastmilk donor bankingʼ on line 218 p.11. Is this a main theme or sub-theme? Is this the same as the significance of breastmilk?

Can you clarify the sub-theme that line 264-282 is addressing? p.13 to 14

I suggest that the result be presented under the main theme. The cultural, ethics and religious considerations theme was presented under benefits of breastmilk in the manuscript. P. 14. This theme should be presented separately.

Clarify the sentence some people believed that given a baby's immunization, it wasn't safe to use it to feed themʼ, line 299-300 p.14

Line 421 to 430, Funding and sustainability were not captured as a sub-theme. I suggest you put it as a sub-theme under challenges and concern p.20

Correct the sentence ʽFrom the planning stage through the implementation stage, education was advised at every stageʼ as the intervention has not been carried out yet. line 481-482, p.22.

The statement ʽinvolve stakeholders in the development of a policy..........ʼ on line 509-510 was not captured as a sub-theme under pre-implementation. p.23.

The result for the themes; opportunity cost and intervention coherence under acceptability were not reported.

Limitations

Report on the limitations of the study

Discussion

Support line 539 -550 with similar or contrary research findings p.25.

Conclude each paragraph with implication or recommendation as you have done in paragraph 6 of the discussion. P.27

Provide references for the sentence ʽThis recommendation was in line with a number of studies......ʼ. Line 614-616, p.28

Include the main contribution of the study to your field of practice or academic knowledge.p. 28

6. PLOS authors have the option to publish the peer review history of their article (what does this mean?). If published, this will include your full peer review and any attached files.

**Do you want your identity to be public for this peer review?** For information about this choice, including consent withdrawal, please see our Privacy Policy.

Reviewer #1: **Yes: **Roberta Maria P Azevedo

Reviewer #2: No

---

## [Editor Report · Decision Letter 1]

6 Jul 2023

PGPH-D-23-00571R1

Acceptability of Breast Milk Donor Banking: A qualitative study among Health Workers in Greater Accra Regional Hospital, Ghana

Dear Authors,

Thank you for submitting your manuscript to PLOS Global Public Health. After careful consideration, we feel that it has merit but does not fully meet PLOS Global Public Health’s publication criteria as it currently stands. Therefore, we invite you to submit a revised version of the manuscript that addresses the points raised during the review process.

We look forward to receiving your revised manuscript.

Kind regards,

Shela Hirani, PhD, IBCLC, RN

Academic Editor
---

## [Editor Report · Decision Letter 2]

25 Jul 2023

Acceptability of Breast Milk Donor Banking: A qualitative study among Health Workers in Greater Accra Regional Hospital, Ghana

PGPH-D-23-00571R2

Dear Authors,

We are pleased to inform you that your manuscript 'Acceptability of Breast Milk Donor Banking: A qualitative study among Health Workers in Greater Accra Regional Hospital, Ghana' has been provisionally accepted for publication in PLOS Global Public Health.

Best regards,

Shela Hirani, PhD, IBCLC, RN

Academic Editor